# ParaPO: Aligning Language Models to Reduce Verbatim Reproduction of Pre-training Data

**Tong Chen**$^{\heartsuit\dagger}$ **Faeze Brahman**$^{\spadesuit}$ **Jiacheng Liu**$^{\heartsuit}$ **Niloofar Mireshghallah**$^{\heartsuit}$ **Weijia Shi**$^{\heartsuit}$
**Pang Wei Koh**$^{\heartsuit\spadesuit}$ **Luke Zettlemoyer**$^{\heartsuit}$ **Hannaneh Hajishirzi**$^{\heartsuit\spadesuit}$
$^{\heartsuit}$University of Washington  $^{\spadesuit}$Allen Institute for Artificial Intelligence

## Abstract

Language models (LMs) can memorize and reproduce segments from their pretraining data verbatim even in non-adversarial settings, raising concerns about copyright, plagiarism, privacy, and creativity. We introduce Paraphrase Preference Optimization (ParaPO), a data synthesis pipeline at post-training that fine-tunes LMs to reduce unintentional regurgitation while preserving their overall utility. ParaPO trains LMs to prefer paraphrased versions of memorized segments over the original verbatim content from the pretraining data. To maintain the ability to recall famous quotations when appropriate, we develop a variant of ParaPO that uses system prompts to control regurgitation behavior. In our evaluation on Llama3.1-8B, ParaPO consistently reduces regurgitation across all tested datasets, achieving a 25.4% reduction on unintentional regurgitation in creative writing, whereas unlearning methods are less effective out of their unlearned domain (with only a 2.3% reduction). On the instruction-tuned Tulu3-8B model, ParaPO combined with system prompting successfully preserves desirable quotation recall while reducing unintentional regurgitation by 27.5% in creative writing when instructed not to regurgitate. In contrast, without ParaPO tuning, prompting the model not to regurgitate produces only a marginal reduction. The code, data, and models are available at ⌨ https://github.com/chentong0/ParaPO.

## 1 Introduction

Language models (LMs) may memorize and verbatim reproduce segments of their pretraining data during generation, a phenomenon called regurgitation (Carlini et al., 2021). While *intentional* reproduction of famous quotations and idioms can be beneficial, *unintentional* regurgitation in open-ended contexts poses significant risks (Lu et al., 2024; Merrill et al., 2024). Such unintentional reproduction diminishes an LM's creative capacity while introducing copyright violations (Henderson et al., 2023), plagiarism concerns (Lee et al., 2023), and privacy issues (Brown et al., 2022).

Mitigating unintentional regurgitation involves two critical challenges: achieving effective reduction and preserving model utility (Wu et al., 2023; Kuo et al., 2025; Suri et al., 2025). Effective reduction is difficult because predicting when and what language models will regurgitate remains challenging (Biderman et al., 2023), while existing approaches using unlearning (Shi et al., 2025) or pre-training data filtering (Min et al., 2024) fail to address unintentional regurgitation beyond their targeted domains. Simultaneously, maintaining utility is essential yet complex, particularly since both unintentional regurgitation and intentional quotation recall emerge from the same underlying verbatim memorization mechanisms (Liu et al., 2024b).

To address these challenges, we propose *Paraphrase Preference Optimization* (ParaPO), a simple yet effective data synthesis pipeline at post-training that uses preference optimization to train models to favor paraphrases over memorized segments. A model fine-tuned with

---

$^{\dagger}$Email: chentong@cs.washington.edu

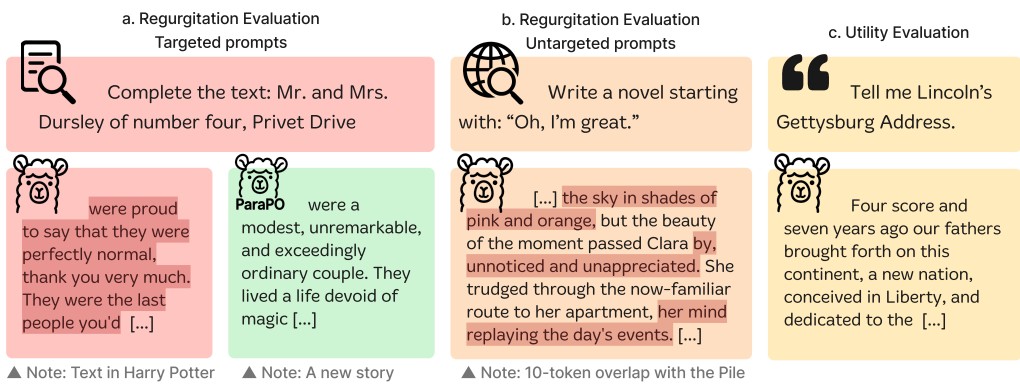

Figure 1: Regurgitation is tested using (a) targeted prompts containing text snippets from known sources and (b) untargeted prompts for creative tasks. Utility is tested using (c) intentional reproduction of famous quotations. Examples are generated by Tulu3-8B with and without ParaPO.

ParaPO on a small training set learns a generalizable ability to reduce verbatim reproduction of pre-training data across diverse prompts. Additionally, we introduce a variant of ParaPO incorporating system prompts that allows for controlled, intentional regurgitation. During inference, the model can be explicitly prompted to either reduce regurgitation or permit it, depending on the instruction provided in the system prompt. To achieve this flexibility, we train the model to prefer memorized text over paraphrased versions when regurgitation is explicitly allowed, and to prefer paraphrased versions when regurgitation is not permitted.

We evaluate ParaPO on both base and instruction-tuned language models across multiple tasks, including targeted and untargeted regurgitation, as well as quotation recall (see Figure 1). On the base model Llama3.1-8B (AI@Meta, 2024), ParaPO reduces regurgitation of book snippets from 15.6 to 1.6 and regurgitation in creative writing from 17.3 to 12.9. These results indicate that the model can internally differentiate memorized content from its paraphrased variants and alter its outputs to reduce verbatim reproduction. In contrast, unlearning methods eliminate regurgitation within the unlearned domain (e.g., reducing book snippet regurgitation from 15.6 to 0.0) but yield little reduction outside of it (e.g., from 17.3 to 16.9 in creative writing). Furthermore, when applied to the instruction-tuned model Tulu3-8B (Lambert et al., 2024), ParaPO combined with system prompts preserves quotation recall accuracy (from 28.0 to 27.5) when permitted to regurgitate, while reducing regurgitation (from 19.9 to 7.6 on web snippets) when prompted not to. These results suggest that the model retains verbatim memory internally but learns to prevent unintentional regurgitation based on instructions.

In summary, we make the following contributions:

1. We propose *Paraphrase Preference Optimization* (ParaPO), a post-training method that reduces regurgitation by training language models to prefer paraphrased content over memorized text. ParaPO enables models to latently distinguish memorized segments from paraphrases and avoid reproducing them.

2. We develop an evaluation framework that jointly measures *unintentional regurgitation* and *intentional quotation recall*, allowing systematic analysis of regurgitation mitigation and utility preservation.

3. We show that ParaPO retains verbatim memory internally while enabling language models to reduce unintentional regurgitation based on system prompts.

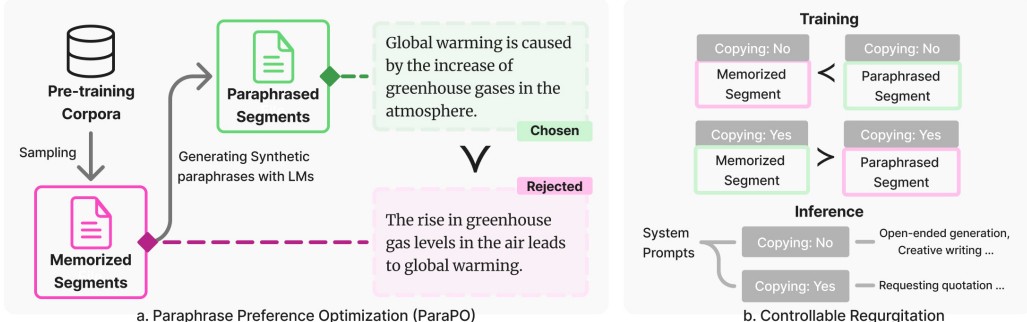

Figure 2: (a) The pipeline for Paraphrase Preference Optimization (ParaPO). We sample memorized segments from the pre-training corpus, generate paraphrases using a strong language model, and apply preference learning to these paraphrase pairs. (b) Controllable regurgitation is introduced using system prompts. We aim to retain the ability to generate verbatim text from the pre-training data, such as recalling quotations.

## 2 Paraphrase Preference Optimization (ParaPO)

In this section, we first introduce our proposed method, Paraphrase Preference Optimization (ParaPO) in §2.1, and describe how ParaPO can be used with system prompts to control regurgitation in §2.2.

### 2.1 Preference Optimization with Paraphrase Pairs

Language models trained with next-token prediction often learn to copy snippets from their pre-training data verbatim. We aim to reduce this copying behavior by guiding the model to lower the probability of memorized segments. The pipeline is demonstrated in Figure 2.

**Constructing Paraphrase Preference Pairs.** To create the training data, we first sample segments from the pre-training corpus and filter out those that are not verbatim memorized. These segments are denoted as $x_1, \ldots, x_n$. We determine whether a segment is verbatim memorized by prompting the target language model with a prefix of the segment and checking whether the model can generate an (almost) exact continuation.

Next, each segment $x_i$ is paraphrased into a corresponding text $y_i$ that preserves the original meaning while using different wording. This process yields the dataset $D = \{(x_i, y_i)\}_{i=1}^n$, which consists of paraphrase pairs.

**Optimization.** We fine-tune the model on $D$ using the Direct Preference Optimization objective (Rafailov et al., 2023, DPO). For each pair $(x_i, y_i)$, the model learns to prefer the paraphrased segment $y_i$ over the segment in the pretaring data $x_i$. Concretely, the loss $\ell(x_i, y_i)$ is given by:

$$- \log \sigma \left( \beta \log \frac{\pi_\theta(y_i)}{\pi_{\text{ref}}(y_i)} - \beta \log \frac{\pi_\theta(x_i)}{\pi_{\text{ref}}(x_i)} \right), \tag{1}$$

where $\pi_\theta$ is the finetuned model's probability distribution, $\pi_{\text{ref}}$ is the reference model's distribution, and $\beta$ is a scaling factor.

### 2.2 Controlling Regurgitation with System Prompts

For instruction-tuned models, we want to enable the use of system prompts to indicate if regurgitation should be reduced in the output. For example, a language model by default can provide exact quotation for queries like "Tell me Lincoln's Gettysburg Address" or "What is the famous beginning of the book A Tale of Two Cities?" In contrast, in some creative tasks, we might want to prompt the language model not to reproduce content verbatim from pretraining data (Chen et al., 2024; Wei et al., 2024; Aerni et al., 2025).

We introduce a variant of ParaPO by inserting system prompts during training. For simplicity, we consider two types of system prompts: those that discourage regurgitation and those that do not. When the system prompt discourages regurgitation, the model treats $x_i$ (the original segment) as the rejected text and $y_i$ (the paraphrased segment) as the chosen text. Conversely, if the system prompt allows the use of verbatim content from pretraining data, $x_i$ is treated as the chosen text and $y_i$ as the rejected text. In this way, the model learns to respond appropriately to the system prompt with respect to the reproduction of memorized segments.

## 3   Experimental Setup

In this section, we describe the training data construction (§3.1), the evaluation benchmarks (§3.2), and the models and baselines (§3.3).

### 3.1   ParaPO Training Dataset Construction

**Source.**   We create training datasets for ParaPO following the process illustrated in Figure 2. We sample segments from The Pile-CC (Gao et al., 2020) dataset, a filtered subset of the Common Crawl[1] dataset, specifically designed for inclusion in The Pile (Gao et al., 2020). We believe recent language models verbatim memorize some of the snippets from the Pile-CC, as recent large-scale pre-training datasets (Soldaini et al., 2024) often use similar sources such as Common Crawl.

**Selection of Memorized Segments.**   We selected memorized segments for each target model to finetune separately. To identify verbatim memorized content, we randomly selected 1 million documents from Pile-CC and evaluated the ability of the target model to generate the exact continuation of a document prefix. Specifically, we prompted the model with the first 64 tokens and requested the generation of the next 32 tokens. We then measured the overlap between the model's output and the actual continuation using ROUGE-L scores. The top 16,000 segments (each consisting of 96 tokens) with the highest ROUGE-L scores were selected to form the training dataset.

**Paraphrases Generation.**   For each selected segment, we generate a paraphrase using Llama3.1-70B-Instruct[2] with a prompt requesting a semantically equivalent segment using different wording. Recalling the preference optimization framework introduced in §2.1, the original segment serves as the "rejected" text (content the model should avoid reproducing verbatim), while the paraphrased version serves as the "chosen" text.

**Controllable Regurgitation.**   As a prototype, we use two fixed system prompts to indicate whether regurgitation should be reduced: Copying: Yes" and Copying: No". In this variant of the training data, we randomly assign one of these system prompts to each paraphrase pair. For paraphrase pairs assigned "Copying: Yes", we reverse the chosen and rejected sequences to encourage verbatim reproduction when it is explicitly specified. Incorporating diverse system prompts is left as future work.

**Data Mixture Composition.**   We also evaluate a variant of the training data that combines ParaPO paraphrase pairs with preference learning pairs, which are previously used to improve general model capabilities. Specifically, we randomly sample a subset from the Tulu3-8B-DPO dataset (Lambert et al., 2024). The ratio of paraphrase pairs to the total training data is treated as a hyperparameter.

### 3.2   Evaluation Benchmarks

**Regurgitation Evaluation.**   We evaluate regurgitation behavior on both targeted evaluation and untargeted evaluation. For targetd evaluation, we evaluate the extractability of these

---

[1] https://commoncrawl.org/
[2] https://huggingface.co/meta-llama/Llama-3.1-70B-Instruct

snippets by prompting the model with the first $N$ tokens from each snippet and comparing its generated continuation (the next $M$ tokens) with the ground-truth reference. To quantify extractability, we define **verbatim overlap** as the ROUGE-L score between the model output and the ground-truth text. This metric computes the longest common subsequence between the two texts, normalized by the length of the reference text ($M$ tokens). We report the **extraction ratio** at half the reference length as the regurgitation score, defined as the percentage of cases in which the ROUGE-L score exceeds 0.5. We use the "Training Data Extraction Challenge" dataset[3] as a web-sourced dataset, and the BookSum dataset used in Wei et al. (2024) as a book-sourced dataset. Note that no snippets are used for training in ParaPO. We use $N = 64$ and $M = 32$ for the web dataset, and $N = 200$, $M = 50$ for the book dataset.

For untargeted evaluation, we use three creative writing tasks: story writing, poem composition, and speech writing. following Lu et al. (2024). To assess the linguistic creativity of generated text, we measure its n-gram overlap with the Pile corpus with Infinite-gram API (Liu et al., 2024a). The **Creativity Index (CI)** quantifies overall linguistic creativity as the average uniqueness across 5- to 11-grams, where n-gram uniqueness is calculated as one minus the n-gram overlap ratio. The **n-gram overlap ratio** represents the fraction of overlapping n-grams in the model output. We report the 11-gram overlap ratio as the regurgitation score. For this metric, we set the sampling temperature to 0.7 and top-p to 0.9.

**Utility Evaluation.** We employ widely used benchmarks from Lambert et al. (2024) to assess different model capabilities: MMLU (Hendrycks et al., 2021) for knowledge retention, GSM8K (Cobbe et al., 2021) for math, and BBH (Suzgun et al., 2023) for logical reasoning. For instruction-tuned models, we additionally evaluate instruction following ability on IFEval (Zeng et al., 2024) and AlapacaEval2 (Li et al., 2023; Dubois et al., 2024; 2023).

For evaluating quotation generation capabilities, we test models on public-domain datasets of books and poems using the BS-NC and BEP datasets from Liu et al. (2024b). We prompt the model to generate the exact content based on a given title. For public-domain books, we use the prompt: `Provide the first 100 words of title by author`. For poems, we use: `Provide the poem title by author`. We quantify accuracy by computing ROUGE-L between the first 50 words of the generated output and the first 50 words of the reference text. Similar to extraction tasks, we report the **quotation recall** as the proportion of cases where the ROUGE-L score is over 0.5.

### 3.3 Models and Baselines

**Models.** We apply ParaPO to both pre-trained base models and instruction-tuned models. For the pre-trained base models, we use LLaMA3.1-8B (AI@Meta, 2024) and Qwen2.5-7B (Qwen-Team, 2024). For the instruction-tuned models, we evaluate both the model tuned only with supervised fine-tuning (Tulu3-8B-SFT) and the model further tuned with DPO and reinforcement learning (Tulu3-8B) and apply ParaPO on these models.

**Baselines.** We evaluate two unlearning methods in our setting: Gradient Ascend (GA) and Negative Preference Optimization (NPO, Zhang et al. (2024)). However, unlearning differs from our method, as it requires specifying an unlearning target dataset, whereas our method reduces verbatim copying in all cases. We use the BookSum dataset, a dataset used in our evaluation, as the forget set. We used the Harry Potter FanWiki dataset as the retain set, following the setup used in the MUSE benchmark (Shi et al., 2025).

We evaluated two alternative post-training methods. First, we fine-tuned the base model on paraphrases of the memorized segments using the continual pre-training objective (denoted as Training on Paraphrases). Second, we applied ParaPO to randomly sampled segments from Pile-CC instead of the memorized segments (denoted as ParaPO w/ Rand Seg).

For the instruction-tuned models, we additionally evaluated the use of system prompts (Chen et al., 2024; Wei et al., 2024; Aerni et al., 2025) to instruct the model not to

---

[3] `https://github.com/google-research/lm-extraction-benchmark`

| Methods | Regurgitation Evaluation (↓) | | | Utility Evaluation (↑) | | | |
|---|---|---|---|---|---|---|---|
| | Web | Book | Creativity | Knowledge (MMLU) | Math (GSM8K) | Reasoning (BBH) | Quote |
| **Llama3.1 8B** | 33.4 | 15.6 | 17.3 | 64.0 | 58.0 | 63.3 | 26.5 |
| + Unlearning Book (GA) | 28.2 | 0.4 | 16.9 | 63.9 | 61.0 | 62.0 | 17.0 |
| + Unlearning Book (NPO) | 28.3 | 0.0 | 17.7 | 64.0 | 60.0 | 62.3 | 17.0 |
| + Training on Paraphrases | 31.9 | 11.8 | 17.8 | 63.9 | 52.5 | 60.7 | 20.0 |
| + ParaPO w/ Rand Seg | 24.4 | 12.6 | 15.2 | 63.7 | 54.5 | 62.4 | 15.5 |
| + ParaPO | 21.6 | 1.6 | 12.9 | 61.2 | 53.5 | 59.8 | 1.5 |
| **Qwen2.5 7B** | 35.3 | 1.8 | 15.1 | 71.9 | 83.5 | 67.1 | 31.0 |
| + Unlearning Book (GA) | 34.3 | 1.6 | 15.8 | 71.6 | 79.5 | 62.9 | 30.5 |
| + Unlearning Book (NPO) | 34.6 | 1.4 | 14.3 | 71.6 | 78.5 | 63.0 | 30.5 |
| + Training on Paraphrases | 35.0 | 1.8 | 15.4 | 72.0 | 82.5 | 21.8 | 32.5 |
| + ParaPO w/ Rand Seg | 33.1 | 1.8 | 12.5 | 70.7 | 84.0 | 68.5 | 26.0 |
| + ParaPO | 29.5 | 0.6 | 10.2 | 70.8 | 86.5 | 68.3 | 12.5 |

Table 1: Regurgitation and utility evaluation of pre-trained base models. ParaPO consistently reduces regurgitation across all tested datasets while maintaining strong utility on MMLU, GSM8K, and BBH.

reproduce the pre-training data verbatim (denoted as System Prompt). We also compare the default ParaPO data with the data variant with system prompts (denoted as Sys) and with the variant that uses a mixture of paraphrase pairs and generic preference data (denoted as Mix). Copy-Y and Copy-N indicate which system prompt is used during inference.

# 4 Main Results

In this section, we aim to demonstrate that LMs trained with ParaPO effectively reduce regurgitation across multiple scenarios while preserving the utility of the models. We present results for pre-trained base models (§4.1) and instruction-tuned models (§4.2).

## 4.1 Pre-trained Base Models

**Unlearning Methods Only Reduce Regurgitation Within the Unlearned Domain.** Table 1 shows the results for Llama3.1-8B and Qwen2.5-7B tuned with unlearning, supervised fine-tuning, and ParaPO. The two baselines, Gradient Ascend (GA) and Negative Preference Optimization (NPO), unlearn the target dataset (BookSum), and thus successfully reduce verbatim reproduction of BookSum to near zero. However, this reduction does not transfer to other domains. The regurgitation scores on web snippets only drop modestly (e.g., from 33.4 to 28.2 (GA) and 28.3 (NPO) on Llama3.1-8B), and regurgitation scores on creative writing remain almost the same. This suggests that unlearning only works on the target unlearned domain and does not affect the general regurgitation behavior of the model.

**ParaPO Effectively Reduces Regurgitation in All Datasets.** ParaPO achieves the largest reduction in copying behavior on all tested datasets. Compared to Llama3.1-8B, regurgitation scores drop from 33.4 to 21.6 (Web), 15.6 to 1.6 (Book), and 17.3 to 12.9 (Creativity Writing). We find that preference learning is necessary because supervised finetuning on paraphrased versions of memorized sequences is less effective in reducing regurgitation.

**Verbatim Memorized Segments are Crucial for Reducing Regurgitation.** Notably, when ParaPO is applied to randomly sampled segments (+ ParaPO w/ Rand Seg) instead of verbatim memorized ones, its effectiveness decreases. In that case, Book extraction score is 12.6, better than baseline, but much worse than 1.6 achieved by the default version of ParaPO. This highlights the importance of using actual verbatim memorized content in the training process. We speculate that the model has difficulty preferring one response over the other in a preference pair unless one version was verbatim memorized and another is not. In such cases, the model learns to assign a lower probability to the memorized text after ParaPO, decreasing the chance of verbatim copying.

| Methods | Regurgitation Evaluation (↓) | | | Utility Evaluation (↑) | | | |
|---|---|---|---|---|---|---|---|
| | Web | Book | Creativity | K & R | IF (IFEval) | IF (AE2) | Quote |
| **Lllama3.1-8B** | 33.4 | 16.0 | 17.3 | 61.8 | / | / | 26.5 |
| + Tulu-SFT | 22.1 | 1.8 | 15.0 | 67.4 | 64.1 | 8.5 | 30.0 |
| + Tulu | 19.9 | 1.2 | 8.7 | 73.9 | 78.2 | 32.8 | 28.0 |
| + Tulu + System Prompt | 16.0 | 1.0 | 8.4 | 73.9 | 78.2 | 32.8 | 17.3 |
| + Tulu + ParaPO w/ Rand Seg | 18.3 | 0.4 | 7.1 | 73.3 | 79.3 | 33.9 | 24.5 |
| + Tulu + ParaPO | 15.7 | 0.0 | 5.5 | 70.9 | 74.3 | 32.6 | 5.5 |
| + Tulu + ParaPO Sys (Copy-N) | 0.1 | 0.0 | 3.7 | 65.0 | 53.2 | 5.2 | 0.0 |
| + Tulu + ParaPO Sys (Copy-Y) | 14.4 | 0.6 | 8.6 | 71.5 | 77.3 | 26.4 | 21.0 |
| + Tulu + ParaPO Mix | 16.7 | 0.2 | 7.1 | 72.9 | 78.2 | 34.1 | 16.5 |
| + Tulu + ParaPO Sys Mix (Copy-N) | 7.6 | 0.6 | 6.3 | 72.5 | 71.9 | 23.9 | 10.5 |
| + Tulu + ParaPO Sys Mix (Copy-Y) | 16.1 | 0.4 | 8.6 | 72.7 | 76.2 | 34.8 | 27.5 |

Table 2: Evaluation of instruction-tuned models on regurgitation and utility. ParaPO with system prompts reduces copying across all domains while maintaining the base model's utility. K&R: Knowledge and Reasoning; IF: Instruction Following; AE2: AlpacaEval2.

Although ParaPO effectively reduces verbatim reproduction of pre-training data, models tuned with ParaPO show slightly lower performance in knowledge, math, and reasoning tasks . We will further discuss how to preserve utility in Section 4.2.

## 4.2 Instruction-Tuned Models

**Generic Instruction Tuning Reduces Regurgitation.** Instruction tuning leads to large reductions in verbatim copying across multiple settings. Starting from the base Llama3.1-8B model, which scores 33.4 on web data and 16.0 on book data in the regurgitation evaluation, applying Tulu-SFT lowers these scores to 22.1 and 1.8, respectively. The full Tulu model (+ Tulu), which incorporates preference optimization and reinforcement learning, further reduces them to 19.9 (web) and 1.2 (book), and lowers the creativity copying score from 17.3 to 8.7. These reductions are achieved even while improving performance on utility metrics.

**ParaPO Further Reduces Regurgitation on Top of The Instruction-tuned Model.** Adding ParaPO on top of the Tulu-3 model reduces regurgitation even further. The Tulu model, further tuned with ParaPO (+ParaPO), eliminates the copying ratio on books, bringing it down to 0.0, and reduces n-gram overlap in creative writing from 8.7 to 5.5. Notably, these improvements in reducing regurgitation come with preserved utility of the model: instruction-following performance, evaluated on AlpacaEval2, remains high at 32.6—nearly matching the base Tulu's 32.8.

**ParaPO Enables Controllable Regurgitation with System Prompts** An inference-time mitigation using a system prompt (+ System Prompt) that reminds the model not to regurgitate has only a limited effect on reducing the copying score: from 19.9 to 16.09 on Web, from 1.2 to 1.0 on Book, and from 8.7 to 8.4 on creative writing. Training the model with ParaPO using system prompts during training and a mixture of generic preference learning data (+ ParaPO Sys Mix) preserves quotation recall ability (28.0 to 27.5) when regurgitation is not discouraged (Copy-Yes), and significantly reduces regurgitation on Web (19.9 to 7.6), Book (1.2 to 0.6), and creative writing (8.7 to 6.3) when regurgitation is discouraged through the system prompt (Copy-No).

**Joint Training on Paraphrases and General Preference Data Further Preserves Utility.** Training the model with paraphrase pairs alone (e.g., +ParaPO and +ParaPO Sys) can lead to a degradation of the original capabilities, especially in instruction-following evaluations, since the quality of long-form generation may be affected by ParaPO training. However, we find that combining ParaPO with both system prompting and human preference data improves the overall balance between low copying and high utility. In ParaPO Sys Mix (Copy-Y), the model achieves an IFeval score of 76.2 and a 34.8 score on AlpacaEval2, retaining the instruction-following ability of the base Tulu model.

| Methods | Story | | | Peom | | | Speech | | |
|---------|-------|--------|---------|------|--------|---------|--------|--------|---------|
| | CI (↑) | 5-gram (↓) | 11-gram (↓) | CI (↑) | 5-gram (↓) | 11-gram (↓) | CI (↑) | 5-gram (↓) | 11-gram (↓) |
| Llama3.1 8B | 0.378 | 99.0 | 17.3 | 0.472 | 96.5 | 12.4 | 0.393 | 97.9 | 22.3 |
| + ParaPO | 0.408 | 98.4 | 14.0 | 0.468 | 97.1 | 11.9 | 0.421 | 97.5 | 13.7 |
| Tulu 3 8B SFT | 0.352 | 98.5 | 19.8 | 0.548 | 95.6 | 7.1 | 0.349 | 98.2 | 23.3 |
| + ParaPO | 0.462 | 98.2 | 7.3 | 0.577 | 96.5 | 4.0 | 0.428 | 98.5 | 10.8 |
| Tulu 3 8B | 0.488 | 97.4 | 7.5 | 0.654 | 92.4 | 1.8 | 0.398 | 98.4 | 16.8 |
| + ParaPO | 0.536 | 97.0 | 3.8 | 0.655 | 95.4 | 1.3 | 0.452 | 98.4 | 10.6 |

Table 3: Creativity Index (CI) and n-gram overlap (5-gram and 11-gram) between language model outputs and the Pile corpus. ParaPO improves CI and reduces n-gram overlap, with a stronger effect on reducing longer (11-gram) verbatim matches than shorter (5-gram) ones.

## 5 Analysis

We further analyze how ParaPO impacts the verbatim reproduction of pre-training data. We discuss the length of verbatim reproduction (§5.1), the probability change of verbatim memorized segments (§5.2), the quality of paraphrases (§5.3), and the case study (§5.4).

### 5.1 Length of Verbatim Reproduction

We compare the n-gram overlap on shortest span (5-gram) with the longest span (11-gram) evaluated in the creative writing dataset. Table 3 presents the creativity index and n-gram overlap ratios across different models and tasks. We observe that most of the improvement of the creativity index comes from a lower overlap ratio in longer spans (11-grams) compared to shorter spans (5-grams). For example, the 11-gram overlap ratio decreased from 16.8 to 12.3 for Tulu3-8B-SFT and from 8.7 to 5.5 for Tulu3-8B. In contrast, the 5-gram overlap ratio remained nearly unchanged across all models. Therefore, we speculate the LM learns to avoid regurgitation by deviating from the memorized text when the model have already generated a long sequence of memorized text.

### 5.2 Probability Change of Memorized Text

In §4.1, we show that using verbatim memorized text during ParaPO training plays an important role in reducing regurgitation. To understand this behavior, we analyze how the model assigns probability to different text snippets. We sample 10,000 documents from the Pile-CC dataset that are not used in training, and use them as a held-out test set. We compute the negative log-likelihood (NLL) for each snippet using both the base Llama3.1-8B model and the ParaPO trained model. The results are shown in Figure 3, where a higher NLL corresponds to a lower assigned probability. Most points lie close to the diagonal line (where NLL before and after training are equal), but many snippets with low pre-training NLL (toward the left side of the plot) exhibit a noticeable increase in NLL after ParaPO training. This indicates that ParaPO decreases the log-likelihood more for text that is likely memorized compared to text that is not.

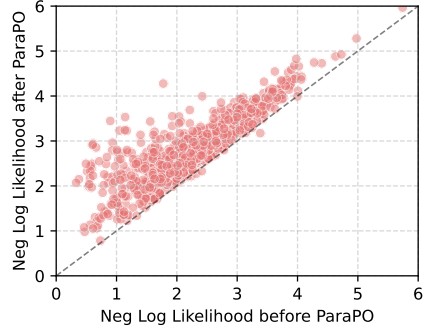

Figure 3: Negative log-likelihood (NLL) of text snippets from the test split of Pile-CC before and after ParaPO training. Snippets with lower NLL before training tend to show a greater increase in NLL after training.

| Methods | Paraphrase Model | Regurgitation Evaluation (↓) | | | Utility Evaluation (↑) | | |
|---|---|---|---|---|---|---|---|
| | | Web | Book | Creativity Writing | Knowledge (MMLU) | Math (GSM8K) | Reasoning (BBH) |
| **Llama3.1 8B** | / | 33.4 | 15.6 | 17.3 | 64.0 | 58.0 | 63.3 |
| + ParaPO | Llama3.1-70B | 21.6 | 1.6 | 12.9 | 61.2 | 53.5 | 59.8 |
| + ParaPO | GPT4.1 | 22.6 | 1.4 | 11.7 | 63.7 | 51.0 | 61.1 |
| + ParaPO | Llama3.1-8B | 18.9 | 0.0 | 18.6 | 62.1 | 46.0 | 55.8 |

Table 4: Training with paraphrases generated by different models.

| Original Text | Paraphrase | Type | Counts |
|---|---|---|---|
| At Vista Clear Eye Care, we strive to provide comprehensive, primary eye care for the whole family. Preventative and routine eye exams are important to maintaining good eye health. | At Vista Clear Eye Care, our mission is to deliver thorough, primary eye care services to families. Regular check-ups and preventative eye exams play a vital role in preserving optimal eye health. | Correct | 14/20 |
| The second one doesn't like right imo, the contrast of a simple 2d image with a 3d background isn't workin for me. The first and last one are nice tho. Can someone explain to me what Android is? I've kinda been living in a cave (not literally) lately. | The second option doesn't seem quite right to me, the juxtaposition of a basic 2D image against a 3D backdrop isn't doing it for me. | Addition or Missing Information | 4/20 |
| Developed to Advance The Finest in Visual Art, Four Points Contemporary Is Holding Its 1st Biannual Juried International Competition and 1st Annual All Media Juried Online Exhibition to Recognize The Best in Undiscovered Talents. | Created to Foster Excellence in Visual Creativity, Four Points Contemporary Presents Its Inaugural Biannual Juried Global Competition and First Annual All Media Juried Online Showcase to Identify Outstanding Emerging Artists. | Terminology Mismatch | 2/20 |

Table 5: Examples of paraphrase classification and associated error .

## 5.3 Quality Analysis of Paraphrases

The effectiveness of ParaPO depends critically on paraphrase quality. We evaluate this dependency by training Llama3.1-8B models using paraphrases generated by three different models: GPT-4.1 (strongest), Llama3.1-70B-Instruct (baseline), and Llama3.1-8B-Instruct (weakest). Table 4 demonstrates a clear correlation between paraphrase model strength and ParaPO effectiveness. Models trained with weak paraphrases (8B-Instruct) show degraded utility and reduced regurgitation benefits. Conversely, high-quality paraphrases (GPT-4.1) match or exceed baseline performance. This occurs because low-quality paraphrases contain surface-level artifacts—poor fluency, unnatural phrasing, or grammatical errors—that enable the DPO algorithm to distinguish text pairs based on linguistic quality rather than memorization patterns, undermining the training objective. To quantify paraphrase quality, we manually evaluated 20 samples from Llama3.1-70B-Instruct. Human inspection revealed two primary failure modes: information distortion (addition or omission of facts) and named entity errors (incorrect person names, locations, etc.). Table 5 provides examples and frequencies. Despite these issues, the 70B model produces sufficiently high-quality paraphrases for effective ParaPO training.

## 5.4 Case Study

We examine how ParaPO modifies next-token predictions at the word level using a memorized sequence from literature. Table 6 shows probability distributions for each token position of the sequence "Mr. and Mrs. Dursley of number four, Privet Drive were proud to say that...". The base Llama3.1-8B model exhibits strong memorization: it predicts the original continuation with high confidence (e.g., "proud" with 91.7% probability at position 17, "say" with 99.8% probability at position 19). After ParaPO training, the model's behavior fundamentally changes. At critical decision points, probability mass redistributes away from memorized tokens toward alternative continuations. For instance, instead of "proud," the model now considers diverse alternatives like "not", "a", and "enjoying" with

| Token Index | 10 | 11 | 12 | 13 | 14 | 15 | 16 | 17 | 18 | 19 | 20 | 21 |
|---|---|---|---|---|---|---|---|---|---|---|---|---|
| Token | _number | _four | , | _Priv | et | _Drive | , | _were | _proud | _to | _say | _that |
| **Next-token Prediction w/ Prob.**

(Llama3.1-8B) | **_four**
(0.975)
_
(0.016)
_Four
(0.006) | ,
(0.884)
_Priv
(0.106)
,C
(0.002) | _Priv
(0.977)
_priv
(0.012)
Priv
(0.006) | et
(0.998)
ett
(0.001)
ets
(0.001) | _Drive
(0.993)
_drive
(0.005)
_Dr
(0.001) | ,
(0.863)
_were
(0.049)
_seemed
(0.02) | _were
(0.955)
_seemed
(0.017)
_had
(0.004) | **_proud**
(0.917)
_happy
(0.046)
_not
(0.015) | _to
(0.997)
_and
(0.001)
_of
(0.001) | **_say**
(0.998)
_see
(0.001)
_be
(0.000) | _that
(0.952)
_they
(0.042)
_the
(0.001) | - |
| **Next-token Prediction w/ Prob.**

(Llama3.1-8B +ParaPO) | _
(0.564)
_four
(0.388)
_Four
(0.022) | ,
(0.978)
_Priv
(0.018)
_priv
(0.000) | _Priv
(0.931)
_priv
(0.025)
_
(0.013) | et
(0.998)
y
(0.000)
ett
(0.000) | _Drive
(0.871)
_Road
(0.032)
_Street
(0.018) | ,
(0.952)
_were
(0.01)
_had
(0.005) | _were
(0.317)
_had
(0.117)
_Sussex
(0.043) | **_not**
(0.051)
_a
(0.042)
_enjoying
(0.031) | _to
(0.267)
,
(0.236)
_and
(0.208) | **_announce**
(0.163)
_be
(0.127)
_have
(0.112) | _that
(0.612)
_they
(0.199)
,
(0.137) | - |

Table 6: Compute the next-token prediction at each position of the following sequence from the opening of *Harry Potter and the Philosopher's Stone*, which Llama3.1-8B has memorized: "Mr. and Mrs. Dursley of number four, Privet Drive were proud to say that...".

more balanced probabilities. This token-level analysis reveals ParaPO's mechanism: the model learns to recognize when it is following a memorized path and actively diverts to a different sequence, breaking verbatim reproduction while maintaining semantic coherence.

# 6 Conclusion

We propose Paraphrase Preference Optimization (ParaPO) as a post-training method to reduce the regurgitation of pre-training data in open-ended generation. Using only 16k preference pairs, ParaPO effectively decreases the extractability of pre-training data and reduces long verbatim copying in creative writing. Furthermore, we demonstrate that regurgitation can be controlled through system prompts in models tuned with a variant of ParaPO that incorporates system prompts during training. We provide a new perspective on mitigating regurgitation through post-training alignment.

## Limitations

While ParaPO significantly reduces verbatim regurgitation, several limitations remain. First, our experiments are limited to models with up to 8B parameters due to computational constraints. Larger models are known to memorize training data more strongly and show higher rates of regurgitation (Carlini et al., 2021), making them both more difficult and more informative for evaluation. Testing ParaPO on larger models would provide a clearer picture of its effectiveness and scalability. Second, although ParaPO reduces exact copying, it does not directly target non-literal memorization, such as reproducing specific events, characters, or stylistic patterns from training data without using the same wording. As a result, ParaPO alone does not fully address copyright or privacy risks in language models. Future work could investigate ways to extend our method to reduce these subtler forms of memorization.

## Ethical Considerations

**Potential for Misuse.** We acknowledge an ethical concern: our method could be misused by model developers to obscure the use of copyright-protected or ethically problematic training data. By reducing verbatim generation, bad actors might attempt to hide evidence of training on content from sources that explicitly disallow AI training use. This risk extends beyond our work to many studies on memorization reduction. Critically, post-training techniques like ParaPO cannot and should not replace responsible data curation or compliance with data usage agreements—as our results show, reducing verbatim outputs does not erase underlying memorization.

**Intended Use and Limitations.** Our method is designed for legitimate scenarios where training data is already legally and ethically sourced, addressing cases where reducing direct copying benefits privacy protection, prevents accidental plagiarism, or respects content creators' interests. All datasets used in our evaluation were publicly available (specifically, samples from the Pile-CC dataset). We strongly emphasize that ethical data sourcing must be the primary consideration before applying any memorization reduction techniques.

**Responsible Deployment.** We promote transparency by making our methodology reproducible through shared code and evaluation protocols. Practitioners should maintain transparency about training data sources, implement auditing procedures against misuse, and consider broader implications of memorization reduction. This transparency enables the research community to better understand both benefits and risks of such techniques, supporting more ethical AI development practices.

## Acknowledgments

We thank members from the UW NLP and UW ML groups for providing helpful feedback. We thank Hamish Ivison, Tianhua Tao, Saumya Malik, Ananya Jha, and Victoria Graf for proofreading the paper draft. PWK is supported by the Singapore National Research Foundation and the National AI Group in the Singapore Ministry of Digital Development and Information under the AI Visiting Professorship Programme (award number AIVP-2024-001), and by the AI2050 program at Schmidt Sciences. This research was partly developed with funding from the Defense Advanced Research Projects Agency's (DARPA) SciFy program (Agreement No. HR00112520300), and NSF IIS-2044660.

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

# A   Related Works

## A.1   Language Model Memorization

Language model memorization (Carlini et al., 2019) occurs when neural networks store and reproduce exact or near-identical sequences from their training data, with verbatim regurgitation manifesting as direct output of these memorized fragments (Carlini et al., 2021). This behavior emerges through standard training procedures, influenced by model scale (larger models exhibit greater memorization), data repetition (frequently encountered sequences are more readily memorized), and task type (factual tasks demonstrate stronger memorization than reasoning tasks) (Carlini et al., 2023; Wang et al., 2025). Training dynamics reveal that improved language modeling capabilities correlate with both enhanced task performance and increased verbatim memorization (Huang et al., 2024; Xie et al., 2025).

Memorization creates several safety concerns: privacy risks when models store personal or sensitive data (Brown et al., 2022), copyright violations from reproducing protected content (Henderson et al., 2023; Chen et al., 2024; Wei et al., 2024), and compromised evaluation validity when models regurgitate test data encountered during pre-training (Bordt et al., 2024). The distributed encoding of high-risk information across model parameters rather than isolated components complicates targeted removal (Huang et al., 2024).

Mitigation strategies include training techniques (e.g., differential privacy that introduces noise during training) (Chua et al., 2024; Hans et al., 2024), data curation approaches that minimize repetition and filter sensitive content (Min et al., 2024), and machine unlearning methods for selective memorization removal (Shi et al., 2025). Prompt engineering and output filtering provide temporary safeguards but fail to address fundamental causes (Chen et al., 2024; Wei et al., 2024; Liu et al., 2024b).

## A.2   Safety Fine-tuning

Safety fine-tuning adapts LMs to align with human values and reduce harmful outputs (Qi et al., 2024). Recent studies identify challenges in balancing safety with utility, addressing cross-lingual disparities, and preventing adversarial misuse or recovery of hazardous capabilities (Shi et al., 2024; Shen et al., 2024; Łucki et al., 2025).

Safety Fine-tuning research on regurgitation reduction has primarily targeted training LMs to refuse harmful or malicious prompts through supervised fine-tuning or alignment techniques (Liu et al., 2024b; Brahman et al., 2024). Our work focuses instead on reducing unintentional regurgitation in benign contexts, such as extending user-provided prefixes or creating original text.

# B  Data Processing Details

**Synthetic Paraphrases.**  To generate paraphrases of memorized segments, we use Llama3.1-70B-Instruct with a temperature of 0.6 and top-p of 0.9. The model is prompted with the following template:

> **Paraphrasing a given snippet**
>
> Your task is to rewrite the given text or code, maintaining the same meaning while using different words. Follow these guidelines:
> - Preserve the original length as closely as possible.
> - Ensure the rewritten version is clear and grammatically (and syntactically) correct.
>
> The text to rewrite is enclosed below:
> —
> {text}
> —
>
> Please provide only the rewritten version without any additional comments or explanations.

**Data Examples**  Some chosen and rejected snippets from the training data of the LLaMA 3.1-8B model are shown in Table 7.

| Chosen | Rejected |
|---|---|
| Question: Is manual intervention required for all users to implement the fix, or must they delete the file themselves? Alternatively, will this issue be resolved automatically for the majority of users—if not, it may deter people from using this product. | Question: Do all users have to apply the fix specifically, or delete the file manually? Or is this going to be sorted out automatically for most people—if not it is true that it will put people off this product. |
| Situated in Nashville, Tennessee, and positioned at the intersection of education, innovation, and healthcare, VUMC is a vibrant community of professionals united by a shared goal of making a profound impact. This is an environment where your skills will be recognized, your understanding broadened, and your capabilities pushed to new heights. This is an environment where your unique perspective — encompassing diverse backgrounds, ideas, experiences, and leadership styles — is actively sought and honored. This is an environment where team members are aware they are integral to something bigger than themselves. | Located in Nashville, Tennessee, and operating at a global crossroads of teaching, discovery and patient care, VUMC is a community of individuals who come to work each day with the simple aim of changing the world. It is a place where your expertise will be valued, your knowledge expanded and your abilities challenged. It is a place where your diversity — of culture, thinking, learning and leading — is sought and celebrated. It is a place where employees know they are part of |

Table 7: Examples of chosen vs. rejected snippets

# C  Model Training Details

We fine-tuned the Llama-3.1-8B and Qwen-2.5-7B model using Direct Preference Optimization (DPO) for 2 epochs with a global batch size of 64, distributed across 4 NVIDIA A100. All models were trained with DeepSpeed Stage 2 and bfloat16 precision. The default learning rate was 5e-7 unless otherwise specified. We used a linear learning rate scheduler with a

warmup ratio of 0.1 and no weight decay. We experimented with different mixture ratios between the paraphrase pairs and generic DPO data by adjusting the proportions to 1.0:0.0, 0.9:0.1, and 0.5:0.5. These ratios were treated as hyperparameters to assess their impact on performance. In our experiments, we used the 0.9:0.1 ratio for the ParaPO Mix variant and the 0.5:0.5 ratio for the ParaPO Sys Mix variant. All training runs were conducted with $\beta = 0.1$.

## D    Baseline Methods

**Supervised Fine-Tuning with Paraphrases.**    We fine-tune the model using the chosen paraphrases from the ParaPO dataset. The model is trained with a learning rate of 1e-4 for two epochs with LoRA (rank 256).

**System Prompts.**    As an alternative mitigation method, we evaluate the effectiveness of system prompts in reducing memorization. The following system prompt is used:

```
Your primary goal is to produce content that is completely original
and free from any existing text on the Internet. This means that if
someone were to take a few words from your explanation and search
for them online, they can not find any matching results.
```

**Unlearning Methods.**    We train models using unlearning algorithms under the same setup as in MUSE (Shi et al., 2025), modifying only the model architectures and hyperparameters as needed. Specifically, we evaluate two methods: Gradient Ascent with KL Regularization (GA-KLR) and Negative Preference Optimization with KL Regularization (NPO-KLR). For each method, we run experiments on two language models: LLaMA-3.1-8B and Qwen-2.5-7B. We use a learning rate of 1e-5 and a global batch size of 32.

## E    Evaluation Details

**Metrics.**    We use the ROUGE-L score, which considers both precision and recall. Precision is defined as the ratio of the longest common subsequence (LCS) length to the output length, while recall is the ratio of the LCS length to the ground truth length. The final ROUGE-L score is computed as the geometric mean of precision and recall. For word-level ROUGE-L, we use `word_tokenize` from the `nltk` library to tokenize sentences into words.

To assess model memorization in copyright-related evaluations, we compute the QA accuracy using an exact string match between the gold answer and the model output.

**Dataset.**    The datasets used in our experiments are licensed under different terms. The Training Data Extraction Challenge is licensed under Apache-2.0, BookMIA under MIT, BookSum under BSD-3-Clause, and the SHIELD code repository under MIT. For the Training Data Extraction Challenge, we tokenize texts using the GPT-Neo 1.3B tokenizer. In copyright evaluation experiments with BookSum and BookMIA, we use the first 50 words of the reference text to compute the ROUGE-L score.

## F    Additional Experimental Results

**Utility Preservation**    To preserve the model's utility, we train the model with a mixture of ParaPO paraphrase pairs and generic preference optimization data. Moreover, we also tune the model with LoRA (Hu et al., 2022), a parameter-efficient finetuning method. The trade-off between utility preservation and regurgitation reduction is shown in Figure 4. We observe that for both full-parameter finetuning and LoRA finetuning, using 90% of paraphrase pairs and 10% of the generic preference optimization data is sufficient to retain most of the model's utility while still yielding a noticeable reduction in regurgitation. We also find that, under the same hyperparameters, LoRA fine-tuning leads to a greater

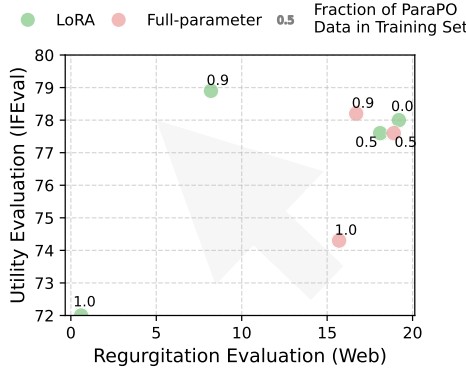

Figure 4: Trade-off between utility and regurgitation when mixing ParaPO with conventional DPO data. Lower paraphrase fractions reduce copying less effectively, while higher fractions preserve utility.

| Model | Methods | Literal (>0.5, %, ↓) | Literal (Max, ↓) | Events (Non-literal) (>4, %, ↓) | Events (Non-literal) (Max, ↓) | Characters (Non-literal) (>2, %, ↓) | Characters (Non-literal) (Max, ↓) |
|---|---|---|---|---|---|---|---|
| Tulu 3 8B | Default | 0.1 | 1.00 | 2.034 | 10 | 5.876 | 7 |
| | ParaPO | 0.0 | 0.46 | 1.243 | 9 | 6.384 | 7 |

Table 8: Literal and non-literal copying results on CopyBench.

reduction in regurgitation. We speculate that full-parameter fine-tuning may reduce the training loss more easily, resulting in smaller overall changes to the model compared to LoRA fine-tuning.

**Full Evaluation on CopyBench.** We evaluate both literal and non-literal copying using CopyBench (Chen et al., 2024), comparing the Tulu-3-8B model with its ParaPO-tuned version. For non-literal copying, we prompt the model with story-writing prompts and measure event and character overlap with reference texts. The results, shown in Table 8, indicate that our method is effective in reducing literal copying but has a limited impact on non-literal copying.

**Evaluation on Enron Email Dataset** Our regurgitation evaluations on web and book snippets directly measure how extractable the training data are from the model, which relates to privacy concerns. We further evaluated the Enron Email dataset (which contains personal information from Enron Corporation employees) and observed the same trend. Using the same setup as the Pile-CC subset, we report the percentage of regurgitated cases in Table 9. ParaPO shows a significant reduction. Manual inspection found that most remaining cases involve website URLs and file paths, which were not changed during paraphrasing in ParaPO.

| | Web (Pile-CC) | Email (Pile-Enron) |
|---|---|---|
| Llama3.1-8B | 33.4 | 4.4 |
| Llama3.1-8B+ParaPO | 21.6 (-35%) | 3.0 (-32%) |

Table 9: Extraction Evaluation on Enron Email.

