# OpenReview forum: "ParaPO: Aligning Language Models to Reduce Verbatim Reproduction of Pre-training Data"
_colmweb.org/COLM/2025/Conference — COLM 2025_

### Official Review · Reviewer_6ZUs · 2025-05-11

**Rating:** 6
**Confidence:** 3
**Ethics Flag:** 2

**Summary:**

This submission describes an attempt to reduce verbatim memorization of language models via preference optimization (DPO) over paraphrased variants (generated from a better language model). The method can be extended to a more controllable variant by using "Copying: Yes" or "Copying: No" as the system prompt.

Many useful experiments have been conducted and the reported results show that the presented methods are in general working as intended.

**Ethics Concerns Details:**

This submission presents a method to discourage verbatim generation of pre-training data. The method could potentially be used to hide copyright infringement from model developers who may unethically use large-scale copyright-protected data for pre-training.

**Questions To Authors:**

1. Why using Llama-3.1 as the paraphrasing model given that many proprietary LLMs might produce better and more diverse paraphrases? Given that the prompt requires to "Preserve the original length as closely as possible", to what extent the rephrasing relies on lexical/phrasal paraphrases versus richer syntactic paraphrases? In what proportion of the paraphrases the semantic meaning was not identical to the original?
2. Any idea on why ParaPO improves Qwen2.5-7B on math and reasoning benchmarks as shown in Table 1?
3. One can read from Table 2 that the controllable variant that discourage verbatim generation (ParaPO + "Copying: No" system prompt) seems to have a negative impact on general utility, especially instruction following (IFEval and AlpacaEval2). What might cause this difference from that without the system prompt?
4. One slightly more convincing reason to disallow verbatim memorization is an intention for privacy preservation. Is there any part of experiments that may reflect such an intention?

**Reasons To Accept:**

- The studied topic (verbatim memorization) might be interesting to many conference attendees.
- The empirical performance seems promising.
- Rich experimental comparisons and analysis have been conducted.

**Reasons To Reject:**

- The presented method is a very straightforward application of DPO on a specific type of data, while the current presentation may leave an impression that ParaPO is a new post-training method.
- Paraphrasing is a key step in this work, while this work currently only includes relevant details in the Appendix without referring in the main text. Current results are based on paraphrases generated by Llama-3.1-70B-Instruct alone, without verification on validity, diversity, or other statistical characteristics of the rephrased texts.
- There has been no discussion at all on a potential misuse: model developers may hide their unethical use of copyright-protected data in pre-training via discouraging verbatim generation. This is a serious concern on many related studies as well, given that many websites have explicitly indicate that they disallow data to be used for training AI models while there's no effective way to technically stop data crawling.
- Doubts in some key results (see questions below)

---

> ### Author Response · Authors · 2025-06-03
> **Reply to Reviewer 6ZUs (Part 1)**
>
> Thank you for your review. We appreciate that you recognize verbatim memorization as an important issue, note the richness of our experiments, and find our proposed method promising. We address your concerns below:
>
> ---
>
> > (R2) Current results are based on paraphrases generated by Llama-3.1-70B-Instruct alone, without verification on validity, diversity, or other statistical characteristics of the rephrased texts.
> > (Q1) [...] Given that the prompt requires to "Preserve the original length as closely as possible", to what extent the rephrasing relies on lexical/phrasal paraphrases versus richer syntactic paraphrases? In what proportion of the paraphrases the semantic meaning was not identical to the original?
>
> Thank you for highlighting the distinction between lexical, phrasal, and syntactic paraphrases. Our dataset contains very few syntactic paraphrases and most are lexical or phrasal. To assess quality, we manually reviewed 20 paraphrases generated by Llama3.1-70B-Instruct and found two main issues: (1) addition or omission of information, and (2) mismatches in named entities. Table A lists examples and frequencies for each issue.
>
> Table A: Quality Analysis of Paraphrases.
> | Original                                                                                                    | Paraphrase                                                                                                 | Type                           | Counts |
> |-------------------------------------------------------------------------------------------------------------|------------------------------------------------------------------------------------------------------------|--------------------------------|--------|
> | [...] Preventative and routine eye exams are important to maintaining good eye health.                      | [...] Regular check-ups and preventative eye exams play a vital role in preserving optimal eye health.      | Correct                        | 14/20  |
> | [...] The second one doesn't like right imo, the contrast of a simple 2d image with a 3d background isn't workin for me. Can someone explain to me what Android is? I've kinda been living in a cave (not literally) lately. | [...] The second option doesn't seem quite right to me, the juxtaposition of a basic 2D image against a 3D backdrop isn't doing it for me. | Addition or Missing Information | 4/20   |
> | [...] Map Of London 32 Boroughs Neighborhoods In is one of good picture from our gallery, you [...]          | [...] London Borough Map 32 Neighborhoods Inside is one of the top images in our collection, you [...]      | Terminology Mismatch           | 2/20   |
>
> ---
>
> > (Q1) Why using Llama-3.1 as the paraphrasing model given that many proprietary LLMs might produce better and more diverse paraphrases? [...]
>
> We agree that paraphrase quality can directly affect preference optimization. To study this, we trained Llama3.1-8B models using paraphrases from GPT-4.1 (stronger), Llama3.1-70B-Instruct, and Llama3.1-8B-Instruct (weaker). Results in Table B show that Llama3.1-8B-Instruct paraphrases lead to worse utility and less regurgitation reduction, while GPT-4.1 paraphrases yield similar or slightly better results compared to Llama3.1-70B-Instruct. If a paraphrase is unnatural or low in fluency, the DPO algorithm may rely on these surface features, reducing its effectiveness on reducing regurgitation.
>
> Table B: Different Paraphrase Models
> | Methods            | Paraphrase Model     | Copying Evaluation (↓): Creativity Writing | Utility Evaluation (↑): AVG of GSM8K, BBH, and MMLU |
> |--------------------|----------------------|--------------------------------------------|------------------------------------------------------|
> | Llama3.1-8B        |                      | 17.3                                       | 61.8                                                 |
> | + ParaPO           | Llama3.1-70B-Instruct         | 12.9                                       | 58.2                                                 |
> | + ParaPO           | GPT4.1               | 11.7                                       | 58.6                                                 |
> | + ParaPO           | Llama3.1-8B-Instruct          | 18.6                                       | 54.6                                                 |

---

> ### Author Response · Authors · 2025-06-03
> **Reply to Reviewer 6ZUs (Part 2)**
>
> > (Q2) Any idea on why ParaPO improves Qwen2.5-7B on math and reasoning benchmarks as shown in Table 1?
>
> We found this behavior surprising as well. One possibility is that DPO may increase the likelihood of Qwen producing better reasoning chains, which could explain the observed improvement in GSM8K and BBH.  Recent work [1] has shown that even RL with incorrect reward can improve Qwen’s performance on math reasoning benchmarks, but this effect appears specific to Qwen and is not observed in models like Llama or Olmo. The authors suggest that the RL objective enhances reasoning paths already encoded in Qwen’s pretraining [1]. Prior work has also shown that DPO training can significantly shift model probability distributions on tasks very different from the training tasks [2].
>
> [1] Shao et al. (2025) Spurious Rewards: Rethinking Training Signals in RLVR
> [2] Ren et al. (2024) Learning Dynamics of LLM Finetuning
>
> ---
>
> > (Q3) One can read from Table 2 that the controllable variant that discourages verbatim generation (ParaPO + "Copying: No" system prompt) seems to have a negative impact on general utility, especially instruction following (IFEval and AlpacaEval2). What might cause this difference from that without the system prompt?
>
> The ParaPO variant with system prompts is a harder optimization problem compared to ParaPO without system prompts. This is because the model must learn to switch behaviors depending on the prompt, which can make it deviate further from the initial checkpoint. In our experiments, we observed that the trade-off between regurgitation reduction and utility preservation is sensitive to the ratio of paraphrase pairs in the training mix, the number of epochs, and the strength of the KL regularizer. Tuning these hyperparameters can recover much of the lost utility, as shown in Figure 4 and Table 2 of our paper.
>
> ---
>
> > (Q4) One slightly more convincing reason to disallow verbatim memorization is an intention for privacy preservation. Is there any part of experiments that may reflect such an intention?
>
> Our regurgitation evaluations on web and book snippets directly measure how extractable the training data are from the model, which relates to privacy concerns. We further evaluated the Enron Email dataset (which contains personal information from Enron Corporation employees) and observed the same trend. Using the same setup as the Pile-CC subset (Lines 145–157), we report the percentage of regurgitated cases in Table C. ParaPO shows a significant reduction. Manual inspection found that most remaining cases involve website URLs and file paths, which were not changed during paraphrasing in ParaPO. Future work can explore methods to address these cases in specific scenarios.
>
>
> Table C: Enron Email
> | Model              | Web (Pile-CC)   | Email (Pile-Enron)  |
> |--------------------|-----------------|---------------------|
> | Llama3.1-8B        | 33.4            | 4.4                 |
> | Llama3.1-8B+ParaPO | 21.6 (-35%)     | 3.0 (-32%)          |
>
> ---
>
> > (Ethics) There has been no discussion at all on a potential misuse: model developers may hide their unethical use of copyright-protected data in pre-training via discouraging verbatim generation. This is a serious concern on many related studies as well, given that many websites have explicitly indicate that they disallow data to be used for training AI models while there's no effective way to technically stop data crawling.
>
> Our method aims to reduce verbatim regurgitation in language models when the training data is already legally and ethically sourced, focusing on scenarios where direct copying is still problematic for privacy, accidental plagiarism, or the interests of content creators. As our paper and prior work show, discouraging verbatim outputs during post-training does not erase underlying memorization. We agree that reducing verbatim generation cannot replace responsible data curation or compliance with data usage agreements. We will clarify these points in our revised ethical considerations.
>
> ---
>
> We hope these clarifications address your concerns, and we welcome any further feedback to strengthen our work.

---

> ### Author Response · Authors · 2025-06-06
>
> We sincerely appreciate the time and effort you have invested in reviewing our work. As we approach the June 10th deadline, we want to ensure we are fully engaged in the discussion process and have addressed each point you raised in our rebuttal. Could you please let us know if you have any follow-up questions, concerns, or would like further clarification? We are glad to continue the discussion and provide any additional information you may need.

---

### Official Review · Reviewer_Jibe · 2025-05-12

**Rating:** 6
**Confidence:** 4
**Ethics Flag:** 1

**Summary:**

This paper introduces ParaPO, a post-training method designed to reduce verbatim reproduction (regurgitation) of pre-training data in language models while preserving their utility. The authors address a key challenge in modern LLMs: their tendency to memorize and regurgitate content from training data, which raises concerns about copyright, plagiarism, privacy, and reduced creativity. Empirical results demonstrate that ParaPO effectively reduces regurgitation across multiple datasets without significantly compromising model utility, outperforming previous unlearning methods that only work on specific target domains. The authors evaluate their approach on both pre-trained base models (Llama3.1-8B, Qwen2.5-7B) and instruction-tuned models (Tulu3 variants), using multiple evaluation metrics. Results show that ParaPO reduces regurgitation across the evaluated datasets while maintaining strong performance on utility benchmarks.

**Questions To Authors:**

1. it's not entirely clear how paraphrase quality impacts the effectiveness of ParaPO. Have you observed cases where poor paraphrases degrade alignment performance? Did you manually or automatically filter any paraphrases for quality?
2. Even though the issue mentioned in the limitation section, since your experiments are conducted on models up to 8B parameters, which are known to memorize less than larger ones, can you speculate on how ParaPO might scale? Is the method expected to remain stable or effective in the context of stronger memorization in larger LLMs?
3. From Table-1, in Math and Reasoning evaluation, applying ParaPo decreases Llama3.1 8B score while increasing the Qwen2.5 7B score. What might be the reason behind this phenomenon?
4. Given the model is explicitly trained to reject verbatim segments and prefer paraphrases, is there a measurable or perceived impact on coherence or discourse structure in long generations? Could this hurt performance on tasks requiring multi-turn or long-span dependencies?

**Reasons To Accept:**

- ParaPO introduces a creative application of preference learning techniques to reduce verbatim copying. While preference optimization has been widely used for aligning models with human preferences, applying it specifically to combat regurgitation is novel.
- Verbatim reproduction is a major concern for language model deployment, with legal, ethical, and practical implications. ParaPO addresses this challenge directly with a practical solution.
- Unlike unlearning methods that only work on specific target domains, ParaPO provides a generalizable approach to reducing regurgitation across different contexts.

**Reasons To Reject:**

- The authors use Llama3.1-70B-Instruct to generate paraphrases. However, there is no analysis of how paraphrase quality affects ParaPO's performance. Understanding whether high-quality paraphrases are necessary or whether simpler rule-based approaches might work would provide valuable insights for practitioners with limited computational resources.
- While the paper shows that ParaPO reduces verbatim copying, the underlying causal mechanism isn't thoroughly explored. For instance, does the model truly learn to paraphrase, or is it simply avoiding certain high-probability continuations? More causal analysis would deepen understanding of how ParaPO works.

---

> ### Author Response · Authors · 2025-06-03
> **Reply to Reviewer Jibe (Part 1)**
>
> Thank you for your review! We appreciate that you find our method novel and practical, and agree that verbatim reproduction is a major concern. We address your suggestions below:
>
> ---
>
> > (R1) The authors use Llama3.1-70B-Instruct to generate paraphrases. However, there is no analysis of how paraphrase quality affects ParaPO's performance.
> > (Q1) it's not entirely clear how paraphrase quality impacts the effectiveness of ParaPO [...]
>
> We agree that paraphrase quality can directly affect preference optimization. To better understand this impact, we are running follow-up experiments using GPT-4.1 (a stronger model) and Llama3.1-8B-Instruct (a weaker model). We trained a Llama3.1-8B model with the same memorized sequence but different paraphrases. As shown in Table A, paraphrasing with Llama3.1-8B-Instruct leads to worse utility and less effective regurgitation reduction, while paraphrasing with GPT-4.1 leads to comparable or slightly better utility and regurgitation reduction compared to Llama3.1-70B-Instruct. Intuitively, if the paraphrase is not natural (for example, if it has low fluency), the DPO algorithm can distinguish between text pairs using these linguistic features rather than true memorization, which results in less effective reduction and lower quality output.
>
> Table A: Different Paraphrase Models
> | Methods            | Paraphrase Model     | Copying Evaluation (↓): Creativity Writing | Utility Evaluation (↑): AVG of GSM8K, BBH, and MMLU |
> |--------------------|----------------------|--------------------------------------------|------------------------------------------------------|
> | Llama3.1-8B        |                      | 17.3                                       | 61.8                                                 |
> | + ParaPO           | Llama3.1-70B-Instruct         | 12.9                                       | 58.2                                                 |
> | + ParaPO           | GPT4.1               | 11.7                                       | 58.6                                                 |
> | + ParaPO           | Llama3.1-8B-Instruct          | 18.6                                       | 54.6                                                 |
>
> ---
>
> > (Q1) [...] Have you observed cases where poor paraphrases degrade alignment performance? Did you manually or automatically filter any paraphrases for quality?
>
> We did not manually filter paraphrases for quality. To check their quality, we conducted a human inspection of 20 paraphrases generated by Llama3.1-70B-Instruct. We identified two main issues: (1) missing or additional information, and (2) mismatches in named entity terminology. Table B provides examples and counts for each case.
>
> Table B: Quality Analysis of Paraphrases.
> | Original                                                                                                    | Paraphrase                                                                                                 | Type                           | Counts |
> |-------------------------------------------------------------------------------------------------------------|------------------------------------------------------------------------------------------------------------|--------------------------------|--------|
> | [...] Preventative and routine eye exams are important to maintaining good eye health.                      | [...] Regular check-ups and preventative eye exams play a vital role in preserving optimal eye health.      | Correct                        | 14/20  |
> | [...] The second one doesn't like right imo, the contrast of a simple 2d image with a 3d background isn't workin for me. Can someone explain to me what Android is? I've kinda been living in a cave (not literally) lately. | [...] The second option doesn't seem quite right to me, the juxtaposition of a basic 2D image against a 3D backdrop isn't doing it for me. | Addition or Missing Information | 4/20   |
> | [...] Map Of London 32 Boroughs Neighborhoods In is one of good picture from our gallery, you [...]          | [...] London Borough Map 32 Neighborhoods Inside is one of the top images in our collection, you [...]      | Terminology Mismatch           | 2/20   |

---

> ### Author Response · Authors · 2025-06-03
> **Reply to Reviewer Jibe (Part 2)**
>
> > (R2) [...] does the model truly learn to paraphrase, or is it simply avoiding certain high-probability continuations? More causal analysis would deepen understanding of how ParaPO works.
>
> We find that ParaPO does not simply avoid high-probability continuations or increase randomness. Instead, it learns to strategically change the next token when it detects a long verbatim match in the prompt. For example, when prompted with the beginning of the first Harry Potter book (“Mr. and Mrs. Dursley of number four, Privet Drive”), the base Llama3.1-8B model generates the exact next line from the book, but the ParaPO model produces a semantically different and non-verbatim continuation (see Figure 1 in the paper). In contrast, when prompted with a factual question (“In natural language processing, the acronym BERT stands for”), both the base model and the ParaPO model correctly answer “Bidirectional Encoder Representations from Transformers.” This shows that ParaPO adapts output depending on context, reducing verbatim memorization while preserving factual knowledge. We will add more examples to the final version.
>
> ---
>
> > (Q2) Even though the issue mentioned in the limitation section, since your experiments are conducted on models up to 8B parameters, which are known to memorize less than larger ones, can you speculate on how ParaPO might scale? Is the method expected to remain stable or effective in the context of stronger memorization in larger LLMs?
>
> Due to computational constraints, we only tested up to 8B parameters.. However, our findings suggest that ParaPO targets strong memorization. In Table C (and Table 2 of the paper), we show that ParaPO reduces not only the percentage of regurgitated test cases but also the maximum length of regurgitated spans (from over 50 tokens to 17). In contrast, the baseline (ParaPO w/ Rand Seg) reduces the average case but still allows long regurgitation in the worst case, consistent with observations in [1]. We expect ParaPO's effectiveness to hold or even improve as model size and memorization increase.
>
> Table C: Maximum Longest Common Subsequence on Book Dataset
> | Methods | Percentage of regurgitated case (ROUGE-L > 0.5) | Maximum length of regurgitated spans (Max LCS) |
> |----------------------------|-------------------|---------|
> | +Tulu | 1.2 | >50 |
> | +Tulu + ParaPO w/ Rand Seg | 0.4 | >50 |
> | +Tulu + ParaPO | 0.0 | 17 |
>
> [1] Wei et al. (2024) Evaluating Copyright Takedown Methods for Language Models
>
> ---
>
> > (Q3) From Table-1, in Math and Reasoning evaluation, applying ParaPo decreases Llama3.1 8B score while increasing the Qwen2.5 7B score. What might be the reason behind this phenomenon?
>
> We found this behavior surprising as well. One possibility is that DPO may increase the likelihood of Qwen producing better reasoning chains, which could explain the observed improvement in GSM8K and BBH.  Recent work [1] has shown that even RL with incorrect reward can improve Qwen’s performance on math reasoning benchmarks, but this effect appears specific to Qwen and is not observed in models like Llama or Olmo. The authors suggest that the RL objective enhances reasoning paths already encoded in Qwen’s pretraining [1]. Prior work has also shown that DPO training can significantly shift model probability distributions on tasks very different from the training tasks [2].
>
> [1] Shao et al. (2025) Spurious Rewards: Rethinking Training Signals in RLVR
>
> [2] Ren et al. (2024) Learning Dynamics of LLM Finetuning
>
> ---
>
> > (Q4) Given the model is explicitly trained to reject verbatim segments and prefer paraphrases, is there a measurable or perceived impact on coherence or discourse structure in long generations?
>
> We evaluated long-form generation quality using two benchmarks: IFEval and AlpacaEval2 (see Lines 167–171). IFEval measures verifiable constraints such as keywords, length, format, and punctuation. AlpacaEval2 uses a language model as a judge to rate overall quality, which implicitly includes coherence and discourse structure. When applying ParaPO & generic instruction tuning data (“+Tulu+ParaPO Mix” in Table 2) to Tulu3-8B (a post-trained version of Llama3.1), both scores are maintained at the same level (IFEval: 78.2 -> 78.2; AlpacaEval2: 32.5 -> 34.1). This suggests that ParaPO does not harm the overall quality or coherence of long-form generations.
>
> ---
>
> Please let us know if there are any remaining questions or concerns. We appreciate your detailed feedback and welcome further suggestions to improve the paper.

---

> ### Author Response · Authors · 2025-06-06
>
> We sincerely appreciate the time and effort you have invested in reviewing our work. As we approach the June 10th deadline, we want to ensure we are fully engaged in the discussion process and have addressed each point you raised in our rebuttal. Could you please let us know if you have any follow-up questions, concerns, or would like further clarification? We are glad to continue the discussion and provide any additional information you may need.

---

### Official Review · Reviewer_nEDD · 2025-05-12

**Rating:** 5
**Confidence:** 3
**Ethics Flag:** 1

**Summary:**

This propose ParaPO to reduce the regurgitation, which is a interesting realistic problem. Specifically, authors build their method on DPO. They first perference data needed by DPO. And the "perference" of ParaPO is the degree of regurgitation. Then, they use these samples to fine-tune LLMs to reduce the regurgitation.

**Reasons To Accept:**

- Clear writing and Detailed presentations: The paper is clearly structured and provides thorough descriptions of both the methodology and evaluation pipelines.
- Controllable regurgitation via prompts: The idea of using system prompts to flexibly control whether regurgitation is desired is simple and practical, enabling LLMs to recall exact quotations when explicitly required.
- New and realistic problems: The authors demonstrate that ParaPO reduces regurgitation metrics across domains (books, web, creative writing) more effectively than baselines like unlearning or simple paraphrase pretraining.

**Reasons To Reject:**

- The contribution is marginal: While the idea is practically useful, the core contribution is essentially applying DPO to a paraphrase dataset constructed for a specific goal (reducing regurgitation). This approach, while effective, does not introduce fundamentally new techniques in optimization or model training.
- The comparison of methods lacks baseline experiments. A significant omission is the lack of comparisons to simple supervised fine-tuning (SFT) baselines trained on the same paraphrase data. Although one such baseline is briefly evaluated (“Training on Paraphrases”), a more rigorous exploration—e.g., combining SFT with system prompts, or training on paraphrase data using next-token prediction instead of DPO—would make the evaluation more complete.
- Insufficient discussion of reinforcement learning (RL) alternatives. Since ParaPO is positioned as a fine-tuning strategy, it would be helpful to compare it against on-policy RL methods (e.g., PPO or GRPO), which are also commonly used to shape model behaviors and could in principle reduce regurgitation. This would strengthen the paper’s claim that ParaPO is superior or more practical.

---

> ### Author Response · Authors · 2025-06-03
> **Reply to Reviewer nEDD**
>
> Thank you for the review! We appreciate that you recognize our writing as clear and detailed, and that our method is simple, practical, and introduces a new and realistic problem for mitigating regurgitation at the post-training stage.
>
> ---
>
> > (R1) The contribution is marginal: While the idea is practically useful, the core contribution is essentially applying DPO to a paraphrase dataset constructed for a specific goal (reducing regurgitation). This approach, while effective, does not introduce fundamentally new techniques in optimization or model training.
>
> We agree that our method builds on established components (e.g., DPO). However, this is the first paper focused specifically on reducing regurgitation in the context of model alignment. Prior work has treated regurgitation mainly as a pretraining or unlearning problem. Our work presents a new and practically useful finding: even when models memorize content during pretraining, they can still be fine-tuned to reduce reproduction without completely removing the knowledge. This reframes regurgitation as a problem that can be mitigated after pretraining, which is realistic for current deployment settings.
>
> ---
>
> > (R2) [...] A significant omission is the lack of comparisons to simple supervised fine-tuning (SFT) baselines trained on the same paraphrase data. Although one such baseline is briefly evaluated (“Training on Paraphrases”), a more rigorous exploration—e.g., combining SFT with system prompts, or training on paraphrase data using next-token prediction instead of DPO—would make the evaluation more complete.
>
> We appreciate this suggestion. Our "Training on Paraphrases" baseline is supervised fine-tuning on the same paraphrase data as ParaPO, and already in the paper (Lines 192–194). To address your suggestion, we conducted an additional experiment combining SFT with system prompts. As shown in Table A, this approach reduces regurgitation compared to the base model, but remains less effective than ParaPO across all evaluations.
>
> Table A: Regurgitation Evaluation
> | Methods                        | Web  | Book  | Creativity |
> |---------------------------------|-------|-------|------------|
> | Llama3.1 8B                     | 33.4  | 15.6  | 17.3       |
> | + SFT (Training on Paraphrases) | 31.9  | 11.8  | 17.8       |
> | + SFT with System Prompts       | 29.3  | 8.4   | 19.4       |
> | + ParaPO                        | 21.6  | 1.6   | 12.9       |
>
> ---
>
> > (R3) Since ParaPO is positioned as a fine-tuning strategy, it would be helpful to compare it against on-policy RL methods (e.g., PPO or GRPO).
>
> We agree that on-policy RL methods such as PPO or GRPO are promising directions for reducing regurgitation. However, designing effective reward functions and constructing training prompts for regurgitation reduction is non-trivial and remains an open question. Our contribution is to show that preference-based fine-tuning (specifically DPO on synthetic paraphrase data) can reduce regurgitation. This result provides a practical alternative to RL-based methods and a foundation for future work that may explore RL approaches for this goal.
>
> ---
>
> Let us know if we have addressed all your questions. We are looking forward to more concrete feedback and to how the paper can be improved.

---

> > ### Comment · Reviewer_nEDD · 2025-06-10
> >
> > Thank you for the clear and thoughtful response. I appreciate the new experiments and agree the paper addresses a relevant and practical problem.
> >
> > However, my main concern remains: the approach builds on existing methods (e.g., DPO) and offers limited novelty in formulation or algorithmic design. The contribution feels more like a new task setup with empirical evaluation than a methodological advance.
> >
> > I will maintain my score, but I appreciate the effort and hope the authors continue developing this line of work.

---

> ### Author Response · Authors · 2025-06-06
>
> We sincerely appreciate the time and effort you have invested in reviewing our work. As we approach the June 10th deadline, we want to ensure we are fully engaged in the discussion process and have addressed each point you raised in our rebuttal. Could you please let us know if you have any follow-up questions, concerns, or would like further clarification? We are glad to continue the discussion and provide any additional information you may need.

---

### Author Response · Authors · 2025-06-10

We sincerely thank all reviewers for their thoughtful feedback and careful evaluation of our work. We are pleased that reviewers recognized the practical value and highlighted the comprehensive empirical evaluation. Below, we summarize key points raised by reviewers and describe our responses and additional experiments.

| | Reviewer nEDD | Reviewer Jibe| Reviewer 6ZUs | Actions/Summary|
|-|-|-|-|-|
| Motivation| NA| "Verbatim reproduction is a major concern for language model deployment, with legal, ethical, and practical implications." | "[...] (verbatim memorization) might be interesting to many conference attendees" | NA |
| Empirical Results | "ParaPO reduces regurgitation metrics [...] more effectively than baselines like unlearning or simple paraphrase pretraining" | "Empirical results demonstrate that ParaPO effectively reduces regurgitation across multiple datasets without significantly compromising model utility"| "Rich experimental comparisons and analysis have been conducted." | NA |
| Preserntation | "The paper is clearly structured and provides thorough descriptions of both the methodology and evaluation pipelines."| NA | "the current presentation may leave an impression that ParaPO is a new post-training method." | Emphasized that ParaPO addresses the novel post-training task of regurgitation reduction rather than introducing a new algorithm. |
| Novelty | "the core contribution is essentially applying DPO to a paraphrase dataset constructed for a specific goal (reducing regurgitation)." | "While preference optimization has been widely used for aligning models with human preferences, applying it specifically to combat regurgitation is novel."| NA| Highlighted that the contribution is applying existing DPO techniques to the problem of unintentional regurgitation for the first time.|
| Paraphrase Quality| NA| "Understanding whether high-quality paraphrases are necessary or whether simpler rule-based approaches might work would provide valuable insights for practitioners with limited computational resources." | "Paraphrasing is a key step in this work, while this work currently only includes relevant details in the Appendix without referring in the main text." | (1) Tested different paraphrase models (GPT-4.1, Llama3.1-70B, Llama3.1-8B). (2) Manual quality assessment.|
| More Study and Discussion | NA| "does the model truly learn to paraphrase, or is it simply avoiding certain high-probability continuations? More causal analysis would deepen understanding of how ParaPO works."| "One slightly more convincing reason to disallow verbatim memorization is an intention for privacy preservation." | (1) Showed ParaPO strategically avoid memorized tokens when detecting verbatim matches while preserving factual knowledge. (2) Tested on Enron Email dataset showing the same trend of improvement.|
| Misc| NA| "in Math and Reasoning evaluation, applying ParaPo decreases Llama3.1 8B score while increasing the Qwen2.5 7B score" and "can you speculate on how ParaPO might scale?" | "why ParaPO improves Qwen2.5-7B on math and reasoning benchmarks" and "(ParaPO + "Copying: No" system prompt) seems to have a negative impact on general utility" | Explained in the individual discussions. |


We thank the reviewers for their detailed and constructive feedback, which has helped us improve the clarity and rigor of our work. We are happy to answer any further questions or discuss additional suggestions.

— Best,

 Authors of Submission 932

---

### Decision · Program_Chairs · 2025-07-08

**Decision:**

Accept

**Comment:**

Mostly positive reviews on this interesting and timely work. I recommend we accept this paper.

The reviewers arguments not to accept the paper include:
- Marginal contribution
- Missing baselines

The reasons to accept include:
- Clear writing
- Important and realistic problem
- Rich experimental findings

Overall, the issue of verbatim regurgitation in LLMs is important and timely. The particular method has several advantages over existing unlearning methods and the empirical results are compelling.

**This paper went through ethics reviewing. Please review the ethics decision and details below.**
Decision: Acceptance (if this paper is accepted) is conditioned on addressing the following in the camera-ready version
Details: A discussion on the potential for dual use should be included.